# Melatonin and Cytokines Modulate Daily Instrumental Activities of Elderly People with SARS-CoV-2 Infection

**DOI:** 10.3390/ijms24108647

**Published:** 2023-05-12

**Authors:** Danielle Cristina Honorio França, Mahmi Fujimori, Adriele Ataídes de Queiroz, Maraísa Delmut Borges, Aníbal Monteiro Magalhães Neto, Phabloo José Venâncio de Camargos, Elton Brito Ribeiro, Eduardo Luzía França, Adenilda Cristina Honorio-França, Danny Laura Gomes Fagundes-Triches

**Affiliations:** 1Biological and Health Sciences Institute, Federal University of Mato Grosso, Barra do Garças 78605-091, Mato Grosso, Brazil; daniellechfranca@gmail.com (D.C.H.F.); mahmi_fujimori@yahoo.com.br (M.F.); adrieleaqueiroz@hotmail.com (A.A.d.Q.); maraisadelmut@gmail.com (M.D.B.); professoranibal@yahoo.com.br (A.M.M.N.); phabloo@outlook.com (P.J.V.d.C.); eduardo.franca@ufmt.br (E.L.F.); dannylauragf@hotmail.com (D.L.G.F.-T.); 2Health Sciences Institute, Federal University of Mato Grosso, Sinop 78557-287, Mato Grosso, Brazil; eltonbr8@gmail.com

**Keywords:** hormone, infection, immunomodulation, aging, acute respiratory syndrome

## Abstract

The Comprehensive Geriatric Assessment analyzes the health and quality of life of the elderly. Basic and instrumental daily activities may be compromised due to neuroimmunoendocrine changes, and studies suggest that possible immunological changes occur during infections in the elderly. Thus, this study aimed to analyze cytokine and melatonin levels in serum and correlate the Comprehensive Geriatric Assessment in elderly patients with SARS-CoV-2 infection. The sample consisted of 73 elderly individuals, 43 of whom were without infection and 30 of whom had positive diagnoses of COVID-19. Blood samples were collected to quantify cytokines by flow cytometry and melatonin by ELISA. In addition, structured and validated questionnaires were applied to assess basic (Katz) and instrumental (Lawton and Brody) activities. There was an increase in IL-6, IL-17, and melatonin in the group of elderly individuals with infection. In addition, a positive correlation was observed between melatonin and IL-6 and IL-17 in elderly patients with SARS-CoV-2 infection. Furthermore, there was a reduction in the score of the Lawton and Brody Scale in the infected elderly. These data suggest that the melatonin hormone and inflammatory cytokines are altered in the serum of the elderly with SARS-CoV-2 infection. In addition, there is a degree of dependence, mainly regarding the performance of daily instrumental activities, in the elderly. The considerable impact on the elderly person’s ability to perform everyday tasks necessary for independent living is an extremely important result, and changes in cytokines and melatonin probably are associated with alterations in these daily activities of the elderly.

## 1. Introduction

The human aging process is marked by the effect of time on the body and is characterized by involvement in physiological, psychological, and social relationships. It is a dynamic process that involves an increase in vulnerabilities, variability in homeostatic balance, and progressive changes in organs and tissues, including brain changes and neurodegenerative progress [1,2]. 

The Comprehensive Geriatric Assessment (CGA) systematizes the multidimensional and interdisciplinary approach to aging and can evaluate dependence parameters in elderly people with different clinical conditions [3,4,5]. CGA is a set of quantitative tests to analyze the aging process, quality of life, and biofunctional and psychological capacity in elderly health [4,5].

In senescence, the physiological aging process, hormonal and immunological changes occur [6,7]. Some studies have proven that the synthesis and secretion of melatonin are reduced in the elderly [8,9]. Melatonin is a hormone identified as an important immunomodulatory agent whose peak production and secretion by the pineal gland occurs at night [10,11]. In addition, in the elderly, sleep disturbances and disorders, such as insomnia, are common [12,13].

The hormone melatonin can be linked to the mobility of immune system cells that act against pathogens [11,14]. In addition, melatonin has been associated with theories of aging, mainly because of the control of the circadian rhythm and its possible anti-inflammatory and immunomodulatory effects [15,16], and the controller of the release of free radicals [14,16].

The aging process begins with theories of damage/programmed errors in the various systems [17], which involve increased inflammation, free radical release, and oxidative stress that occurs when Reactive Oxygen Species are produced [16,18]. In the central nervous system, aging promotes changes in the conformation of neurons, mainly at the dendritic level [19,20], with the brain being the organ most affected by free radicals [21]. Furthermore, aging is associated with cognitive and motor deficits [22,23].

Immunosenescence causes immune dysfunction, changes in pathogen recognition, and genetic alterations with chronic antigenic stimulus, which reduces immunosurveillance and increases the risk of developing infectious processes [7,24] which directly interfere with daily activities, including the ability of the elderly to perform more cognitive brain-based skills, and also affect the response of the elderly to infections [25]. Some respiratory infections have been associated with high lethality in the elderly [26]. COVID-19 “coronavirus disease 2019” is a pathology that courses with a severe acute respiratory syndrome caused by the severe acute respiratory syndrome coronavirus 2 (SARS-CoV-2), an emergent infection that affected a large part of the elderly population [27]. The pandemic has significantly impacted communities worldwide, resulting in numerous deaths and economic and social consequences. Research has also shown that immune and hormonal changes can impact the severity of symptoms and prognosis in older adults [28].

Alterations in the elderly cytokine profile, such as an increase in IL-6 and changes in other pro-inflammatory cytokines, were reported [29], as were correlations between the hormones on immune system cells in viral diseases that cause respiratory symptoms, such as SARS-CoV-1, Middle East respiratory syndrome coronavirus (MERS), and influenza [4,30]. 

During flu syndromes, the organism experiences fatigue, tiredness, and sleep during the day, which characterizes the “sick behavior,” where the patient loses the ability to have his body regulated properly by the sleep–wake cycle, altering homeostasis [31,32]. It is also verified that patients experiencing these infectious processes have abnormalities in the synthesis of melatonin in a rhythmic way [33]. 

Moreover, studies indicate an increase in the rate of severe acute respiratory syndrome in the elderly, especially if they have comorbidities [28]. In addition, an experimental study showed a relationship between cognitive dysfunction and post-infection by the SARS-CoV-2 virus [34]. Thus, the present aim of this study was to analyze the cytokines levels and the melatonin concentration in serum and their relation with the Comprehensive Geriatric Assessment in elderly patients with SARS-CoV-2 infection.

## 2. Results

### 2.1. Clinical Data of Subjects

General characteristics and clinical data of the elderly are described in Table 1. The age of elderly patients was 68.2 ± 7.9 and 68.4 ± 8.0 for the elderly who were non-infected and infected with SARS-CoV-2, respectively. Of the 43 patients in the control group, 25 (58.2%) were female and 18 (41.8%) were male. In the group with COVID-19, which consisted of 30 patients, 16 (52.3%) were female and 14 (46.7%) were male. The groups’ body weight, Body Mass Index, and glucose were similar. Of the 73 elderly individuals evaluated, only 7 were hypertensive, all of whom had controlled hypertension. Most of the elderly lived with their partner or were accompanied by elderly care. None of the participants in this study lived in long-term care or were homeless.

### 2.2. Melatonin and Cytokines Levels

The melatonin level was higher in elderly patients with SARS-CoV-2 infection compared to non-infected individuals (Figure 1).

The cytokines levels in serum from elderly patients infected or not with SARS-CoV-2 are shown in Table 2. The IL-6 and IL-17 were increased in serum from elderly patients infected with SARS-CoV-2 compared to elderly individuals who were not infected. However, the IL-10 concentration was similar between the groups. The Fluorescence Intensity of the cytokines in serum from elderly individuals infected or not with SARS-CoV-2 is represented in Figure 2.

The melatonin and cytokines levels were correlated. It was noticed that there was a positive correlation between melatonin and IL-6 and between melatonin and IL-17 (Table 3).

### 2.3. Comprehensive Geriatric Assessment

Figure 3 shows the basic activities of daily living by the Katz Index. There was no statistical difference, independently of infection, in the basic activities of daily living among the elderly people evaluated (*p* > 0.05). 

Figure 4 shows the analysis of the instrumental activities of daily living of elderly individuals infected or not with SARS-CoV-2 by the Lawton and Brody scale. A reduction in the score of the Lawton and Brody Scale in the infected elderly was observed.

The analysis of the frequency of daily activities of an instrumental nature revealed that 85% of the elderly with COVID-19 showed partial dependence. Only 35% of infected elderly people reported being able to prepare meals, while the rate for the uninfected elderly group was 70%. In addition, 60% of the elderly with infection and 47.5% of the elderly without infection reported being unable to perform heavy tasks. Count money ability was reported by 95% of the elderly without infection and 80% of the elderly with COVID-19. Only medication use without help was higher in the infected group; the rate was 82.5% for elderly people without infection and 90% for elderly people with infection. Shower and dressing alone, unassisted toilet use, mobility, and feeding without help, independently of infection, were above 90% (Table 4).

The schematic representation summarizing the main findings in elderly seropositive for SARS-CoV-2 is illustrated in Figure 5.

## 3. Discussion

The aging process is marked by the effect of time on the human body [1,2]. Therefore, it is crucial to be able to perform basic and instrumental activities of daily living, even when pathologies are present. This not only impacts the immediate disease process, but also affects the overall prognosis post-infection. Then, it is important to maintain a sense of independence and capability as much as possible during recovery [25]. In addition, during its development, immunomodulatory variations occur, such as changes in cytokines and hormones that affect the immune response against pathogens [7]. 

Melatonin is the hormone primarily produced by the pineal gland [35], related to the circadian cycle [36]. The pineal gland and its involvement in inflammatory responses has been studied [11,37]. Its concentration is directly linked to diseases in the elderly as a neuromodulatory biomarker [38]. Calcification of the pineal gland has been studied in the aging process and may be related to this change in melatonin production [39,40]. In addition, melatonin has been identified as a possible reducer of SARS-CoV-2 infection and other respiratory diseases due to its ability to act in different organs and immune system cells [41,42]. 

The use of melatonin against COVID-19 has been a proposed therapy [27,43,44]. However, in the present study, it was possible to notice an increase in melatonin levels in the serum of individuals older than 60 years infected with SARS-CoV-2, which suggests an attempt at negative feedback from the organism itself. 

With the aging process, it is possible to perceive a decrease in several immune system components, such as the reduction in the functional capacity of NK cells and the cellular response mediated by T lymphocytes [45,46]. These alterations explain the increased vulnerability and greater susceptibility of the elderly to acquire pathologies, such as those of an infectious nature [47]. It is also possible to notice an increase in pro-inflammatory cytokines [48].

IL-6 is a cytokine of innate immunity that has a pro-inflammatory action, is important in mediating the production of neutrophils and acute-phase proteins, and is also involved in several pathological processes related to aging, such as Alzheimer’s disease, other neurological disabilities, osteoporosis, and cardiovascular diseases [49,50]. In addition, the increase in this cytokine in elderly people may mean a predisposition to these diseases [51].

According to Li et al. [52], IL-6 is associated with severe clinical conditions of COVID-19, and this cytokine is considered a potential biomarker for the disease. Treatments with antagonists of this cytokine have been tested and demonstrated promising effects against the pathology [53].

This study observed that IL-6 increased in individuals with SARS-CoV-2 infection compared to non-infected individuals. These findings corroborate those of Laguna-Goya et al. [54], who conducted a study involving 501 patients aged 44 to 60 years old who were positive for SARS-CoV-2, and also found a considerable increase in IL-6 and correlated the presence of this biomarker with the severity of clinical conditions. Another study [55] involving infected patients also observed increased IL-6 levels. 

This work also evaluated IL-10, a cytokine with anti-inflammatory action which acts by inhibiting other cytokines of a pro-inflammatory nature, promoting less exacerbated immune responses, maintaining homeostatic processes, and preventing tissue damage [56]. However, the action of IL-10 against the induction of collateral tissue damage can inhibit the activity of T lymphocytes, NK cells, and macrophages, which can compromise the elimination of the pathogen in infectious processes caused by viruses [57].

Because of this mechanism, studies suggest that the balance between IL-6 and IL-10 cytokines should be stimulated for greater control of severe clinical conditions [58]. In addition, in COVID-19, levels of interleukins 6 and 10 are identified as predictors for recognizing high-risk patients [59].

Studies suggest that serological levels of IL-10 are elevated in severe cases of COVID-19 through a negative feedback mechanism suggesting both pro-inflammatory activity and anti-inflammatory response [59,60].

However, a study in the literature showed that compared to other diseases that cause cytokine-releasing syndromes, IL-10 levels in COVID-19 are lower [61]. In aging, IL-10 production by B lymphocytes is reduced [62]. The physiological mechanism of IL-10 in aging still needs to be fully understood. However, the increase of this cytokine in experimental studies with rats of an advanced age on other pathologies showed a beneficial effect in suppressing inflammation [63]. In the present study, IL-10 levels, regardless of the presence of the infection group, were similar between groups. It is important to emphasize that all analyses of this work on individuals positive for SARS-CoV-2 infection were performed on participants with mild to moderate clinical conditions. None of these participants were hospitalized in the intensive care unit at the time of collection.

With aging, the process called inflammaging (aging with inflammation) is associated with increased pro-inflammatory cytokines. IL-17 is a cytokine very present in viral pathologies [64]. Studies have been developed relating IL-17 and its importance in individuals of older age [65], as well as in individuals with comorbidities, such as diabetes and obesity [66]. Moreover, IL-17 can also act on macrophages and increase the production of other cytokines [67,68]. 

In this work, IL-17 was higher in serum from the elderly with SARS-CoV-2 infection. IL-17 has been studied in COVID-19, indicating that there may be a connection between the severity and progression of the disease with the increase in this cytokine. Thus, this has been identified as a potential biomarker for tracking the progression of disease prognostic monitoring for better suggestions for treatments according to the clinical evolution. It has also been described as an amplifier of the immune response in pathology [69]. It can be involved in cases of hyperinflammatory syndrome, with the therapeutic use of anti-IL-17 being proposed in studies [70]. IL-17A has been suggested as the main cytokine responsible for respiratory distress syndrome in COVID-19 during the cytokine storm [71]. In the present study, all participants were tested around the 3rd to 7th day of infection, an adequate interval for RT-PCR according to local protocols [72], suggesting that cytokines can be important in the initial disease. Several studies have reported correlations between cytokines, hormones, and chronic degenerative diseases, including obesity and diabetes [66,73,74,75].

Regarding infection by SARS-CoV-2, it was possible to assess that the role of melatonin can be related to the control of inflammation, since there was a correlation between the increase in this hormone and the increase in IL-6. These data suggest that during infection, the immune response determines an increase in cytokines, especially those with pro-inflammatory activity. Moreover, there is an increase in serum melatonin as a possible feedback mechanism of the human physiological system itself. 

Studies show that elevated cytokines, such as IL-6, can cause impairment and cognitive decline in the elderly [76,77]. Therefore, the Comprehensive Geriatric Assessment (CGA), using the Katz Index and the Lawton and Brody Scale, is used for analyzing the neurocognitive and physical states of the elderly.

Several diseases and their consequences can be monitored using the CGA, which makes it possible to assess the ability of the elderly to perform basic and instrumental daily activities. Examples of basic activities include eating habits, hygiene, and sphincter control [78]. As instrumental activities, the habits of using the telephone, driving, preparing meals, and other activities of a more refined cognitive nature are considered [79]. 

In this study, no significant changes in basic daily activities were found. However, it is essential to highlight that the ability of the elderly to carry out daily instrumental activities was reduced.

Despite the basic activities not showing differences between the groups (elderly with and without positive infection for the SARS-CoV-2 virus), elderly individuals positive for COVID-19 showed less independence in activities of a more refined nature, and less ability to carry out instrumental activities of daily life of a more refined cognitive nature. These findings indicate the need for greater attention on the part of new public policies for disease prevention in this population, and the need for the long-term follow-up of elderly people affected by the infection, even after successful treatment. Furthermore, these changes may indicate impairment in the future daily habits of the elderly who tested positive for COVID-19.

An important result presented in this study was the ability of elderly people seropositive for SARS-CoV-2 to use medication without help, which was higher when compared to those negative for the disease. This can indicate that the infection can make the elderly more attentive to therapeutic practice, because medication adherence is usually low in elderly people affected by diseases, contrary to what was evaluated in this study. In Brazil, studies report adherence rates ranging from 26.7% to 43.3% of elderly people to the use of medicines in general [80,81,82].

## 4. Materials and Methods

### 4.1. Study Design and Participants

This is an analytical, quantitative study with a cross-sectional approach. Based on the sample calculation, for a significance level of 95% and a sampling power of 97.2%, it was recommended to have at least 28 individuals in each group to analyze all variables. The research was developed in Barra do Garças, Mato Grosso, Brazil. Since the research was developed in 2020 and 2021, all participants were submitted to the COVID test real-time reverse transcription–polymerase chain reaction (RT-PCR). Out of the 116 patients who were examined, 28 were not included due to pre-existing conditions such as diabetes, obesity, or autoimmune diseases. Moreover, 15 patients who were under the age of 60 were also excluded. According to serological analysis, the individuals were separated into two groups: Control Group (CG) individuals without infection who are 60 years or older, and Infected Group (IG) individuals with a positive diagnosis of SARS-CoV-2 who are 60 years or older. Eligible participants voluntarily agreed to participate in the study. The sample consisted of 30 participants with SARS-CoV-2 infection and 43 without infection, totalizing 73 participants. The scheme for obtaining samples and the experimental design is described in Figure 6. All patients signed the Informed Consent Form. 

### 4.2. The Comprehensive Geriatric Assessment

Questionnaires were applied for data collection about the Comprehensive Geriatric Assessment: the basic activities of daily living, such as taking a shower, dressing alone, using the toilet without assistance, mobility, and feeding without assistance; the control of sphincters by the Katz Index [78]; and the instrumental activities of daily living by Lawton and Brody, such as the ability to prepare meals, perform heavy tasks, ability to count money, use of medications without assistance, use of the telephone without assistance, traveling alone, and shopping [79]. 

### 4.3. Blood and Serum Separation

In addition, blood samples were collected from the groups by venipuncture (approximately 5 mL) from individuals who attended the Health Services of Barra do Garças-MT during the project development period. Blood samples were centrifuged at room temperature for 15 min at 160× *g*. The serum was separated and stored at −80 °C for posterior cytokines and melatonin assays. 

### 4.4. Glucose Determination

Samples of 20 μL serum, the standard of 100 mg/dL (BioTécnica^®^, Ref 10.008.00, Brazil), were placed in 2.0 mL phosphate buffer solution (0.05 M, pH 7.45, with amino antipyrine 0.03 mM, 15 mM sodium p-hydroxybenzoate, 12 kU/L glucose oxidase and 0.8 kU/L peroxidase). The suspensions were mixed and incubated for 5 min at 37 °C. The reactions were read on a spectrophotometer at 510 nm.

### 4.5. Melatonin Determination

Quantitative determination of melatonin hormone concentration in human serum was performed by a commercial ELISA kit (IBL, Hamburg, Germany), with the following characteristics: the lower detection limit was 1.6 pg/mL, and intra-assay and inter-assay coefficients of variation (%) were 3.0–11.4 and 6.4–19.3, respectively. The melatonin present in the serum was extracted by affinity chromatography. Column preparation followed the protocol established by the manufacturers. They were placed in glass tubes and washed twice with 1 mL of methanol (1 min—200× *g*). Afterward, they were washed twice with bidistilled water (1 min—200× *g*). Then, 0.5 mL of standards, controls, and samples were applied to the column and centrifuged for 1 min at 200× *g*. Then, the columns were washed again with 1.0 mL of 10% methanol for 1 min at 500× *g*. To extract the eluate containing the hormone melatonin, 1.0 mL of methanol was added at 200× *g*. After obtaining the eluate, the methanol was evaporated in a centrifuge evaporator (speed-vac). The material was reconstituted with 0.15 mL of bidistilled water under agitation for 1 min and immediately analyzed. 50 mL of each standard, control, colostrum, and milk sample were placed on an ELISA plate with 50 mL of melatonin-biotin in each well, with 50 mL of antiserum. The plate was incubated at 4 °C for 20 h. After this period, the liquid was discarded, the plate was washed 3 times with washing buffer, 150 mL of the conjugated enzyme was added, and the plate was incubated at room temperature for 120 min. After this period, the plate was again washed 3 times. Next, 200 mL of substrate p-nitrophenyl phosphatase (PNPP) was added and incubated for another 40 min under agitation. The reaction was blocked by 50 mL of “PNPP stop” solution. The reaction values were measured by absorbance in a spectrophotometer with a 405 nm filter. The results were calculated using the standard curve (R² = 0.984) and expressed in pg/mL.

### 4.6. Cytokines Determination

The concentrations of cytokines IL-6, IL-10, and Il-17 present in the serum samples were evaluated using the “Cytometric Bead Array” Kit (CBA, BD Bioscience, San Jose, CA, USA). Three bead populations with different fluorescence intensities were conjugated with a specific capture antibody for each cytokine, mixed to form the CBA, and read in the FL3 channel of the FACScalibur flow cytometer [BD Biosciences^®^, San Jose, CA, USA]. Bead populations were visualized according to their respective fluorescence intensities: from least bright to brightest. In the CBA, the cytokine capture beads were mixed with the detection antibody conjugated to the fluorochrome PE and then incubated with the samples to form the “sandwich” assay. The acquisition tubes were prepared with 50 μL of the sample, 50 μL of bead mixture, and 50 μL of detection reagent (PE Detection Reagent/1 vial, 4 mL). The same procedure was performed to obtain the standard curve. The tubes were incubated for three hours at room temperature without light. These cytokines were analyzed using flow cytometry (FACSCalibur, BD Bioscience, San Jose, CA, USA). The results were generated in graphs and tables using the CellQuest (BD)^®^ software, and the data were analyzed using the FCAP Array software (BD Bioscience, San Jose, CA, USA).

### 4.7. Statistics

Statistical analyses were performed using the BioEstat 5.0 program. Qualitative variables were examined through relative and absolute frequency. Quantitative variables were described using their means and standard deviation. Quantitative data were analyzed for normal distribution using the Shapiro–Wilk test. When necessary, the student’s t-test was used to analyze parametric quantitative variables. For non-parametric quantitative variables, the Mann–Whitney test was performed. For the correlation analysis, the Pearson test was used. The adopted significance criterion was 5% (*p* < 0.05).

## 5. Conclusions

The data from the present study allow us to conclude that there was a reduction in the ability to perform instrumental activities of daily living in elderly people with SARS-CoV-2 infection, suggesting an impact and impairment in the biopsychosocial context. The systemic inflammatory environment determined by the increase in IL-6 and IL-17 cytokines can contribute to these alterations. It also highlights the increase in the hormone melatonin in the serum of these patients, which can be an important feedback mechanism for controlling inflammation during infection for the re-establishment of homeostatic processes. Due to the anti-inflammatory action of melatonin, this hormone can also be a future therapeutic alternative in elderly people with SARS-CoV-2 infection.

## Figures and Tables

**Figure 1 ijms-24-08647-f001:**
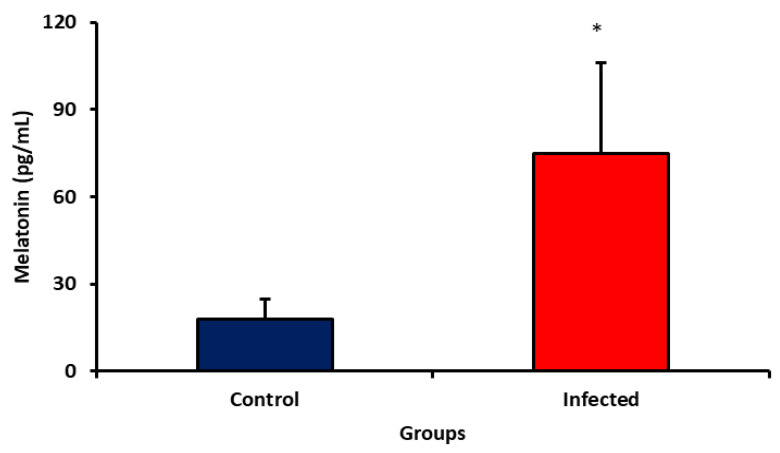
Melatonin concentration in serum from patients with SARS-CoV-2 infection. * *p* < 0.05 significative difference between groups.

**Figure 2 ijms-24-08647-f002:**
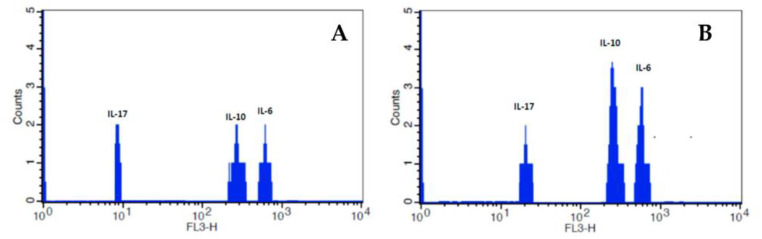
Fluorescence Intensity of cytokines in serum from elderly individuals without infection (**A**) and elderly individuals infected with SARS-CoV-2 (**B**). Fluorescence analyses were carried out by flow cytometry (FACSCalibur, Becton Dickinson, USA). FL3 (Fluorescence in channel 3). IL—Interleukin.

**Figure 3 ijms-24-08647-f003:**
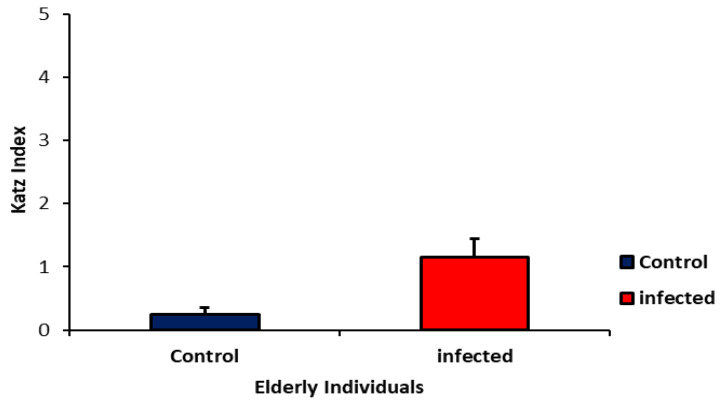
Comprehensive Geriatric Assessment—Daily activities of elderly individuals with and without SARS-CoV-2 virus. Results are presented as mean and standard deviation. *p* > 0.05.

**Figure 4 ijms-24-08647-f004:**
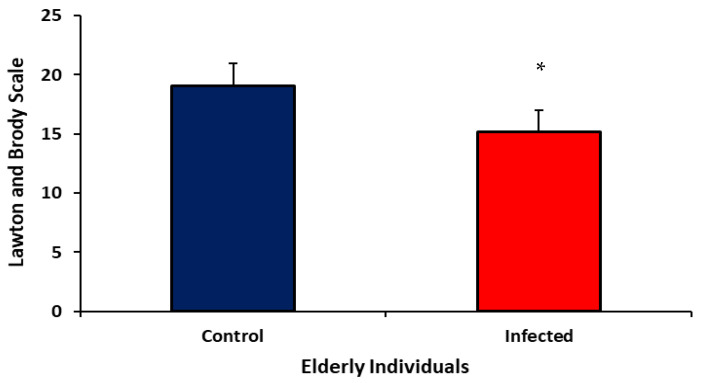
Comprehensive Geriatric Assessment—Instrumental activities of elderly individuals infected or not with SARS-CoV-2 virus. Results are presented as mean and standard deviation. * *p* = 0.038.

**Figure 5 ijms-24-08647-f005:**
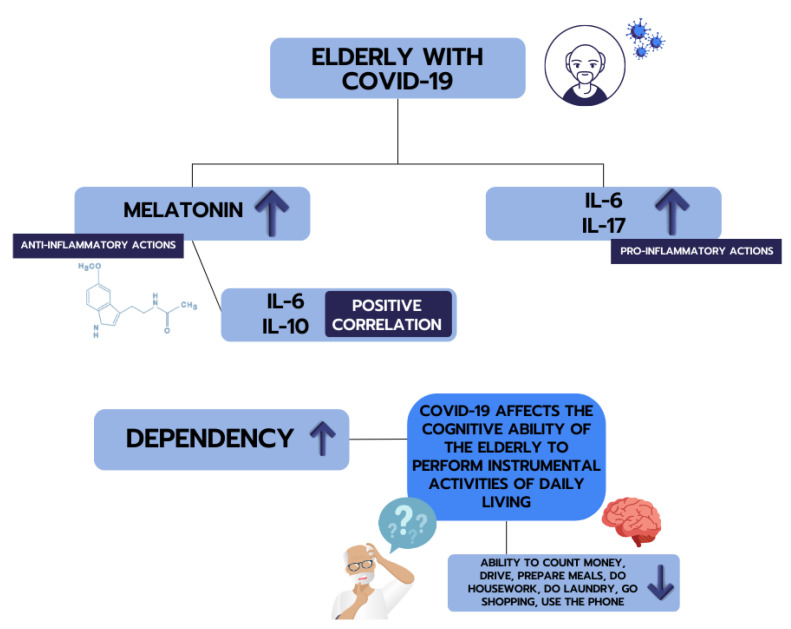
The schematic representation summarizes the main findings in elderly seropositive for SARS-CoV-2. The melatonin, cytokines, and correlations between these immunomodulators and the Comprehensive Geriatric Assessment (CGA) are presented.

**Figure 6 ijms-24-08647-f006:**
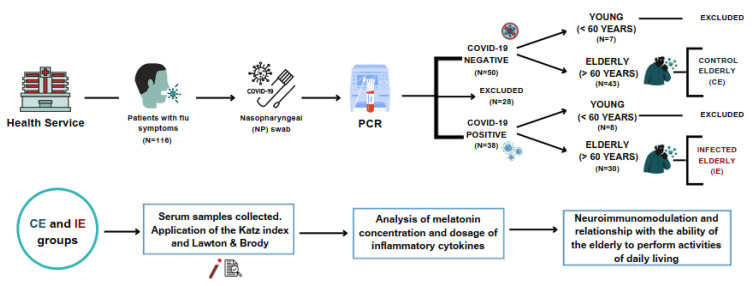
Representative scheme for obtaining samples and experimental design.

**Table 1 ijms-24-08647-t001:** General characteristics and clinical data on patients infected or not infected with SARS-CoV-2.

Parameters	Non-Infected	Infected
	(N = 43)	(N = 30)
Age (years)	68.2 ± 7.9	68.4 ± 8.0
Gender Male Female	18 (41.8%) 25 (58.2%)	14 (46.7%) 16 (52.3%)
Body Mass Index (kg/m^2^)	23.1 ± 1.0	24.3 ± 1.3
Body weight (kg)	57.0 ± 6.7	67.3 ± 8.2
Glucose (mg/dL)	73.1 ± 3.3	88.0 ± 4.5
Diabetes mellitus	0/43 (0.0%)	0/30 (0.0%)
Hypertension (%)	4/43 (9.3%)	2/30 (9.0%)
Smoking (%)	0/43 (0.0%)	0/30 (0.0%)
Hours of Sleep/Night	7.6 ± 0.9	7.2 ± 0.8
Residence:		
Alone	11 (26.0%)	6 (20.0%)
Partner	16 (37.0%)	13 (43.0%)
Family	3 (7.0%)	2 (7.0%)
Caregiver	13 (30.0%)	9 (30.0%)
Long-Term Care Institutions	0	0
Homeless	0	0

Note: The date of age, body mass index, body weight, and glucose are mean and derivation standard; the gender, diabetes mellitus, hypertension, smoking, hours of sleep/night, and home are represented by number and percentage.

**Table 2 ijms-24-08647-t002:** Cytokines level in serum from elderly patients infected with SARS-CoV-2.

Groups	Control	Infected	Statistical
IL-6	5.6 ± 1.7	10.3 ± 1.8 *	t = 6.1488 *p* = 0.0001
IL-10	6.9 ± 2.9	7.9 ± 3.9	t = 0.0716 *p* = 0.4721
IL-17	421.5 ± 70.2	508.0 ± 77.5 *	t = 2.5314 *p* = 0.0309

Data are presented in mean and standard deviation. * statistical difference.

**Table 3 ijms-24-08647-t003:** Correlation between melatonin and cytokines in control and infected groups.

Groups	Control	Infected
MLT-IL-6	r = −0.1947 *p* = 0.5442	r = 0.5474 *p* = 0.0346
MTL-IL-10	r = 0.0035 *p* = 0.9913	r = 0.5805 *p* = 0.0232
MLT-IL-17	r = −0.2417 *p* = 0.4492	r = −0.0638 *p* = 0.8284
IL-6-IL-10	r = 0.4806 *p* = 0.1136	r = 0.2683 *p* = 0.3335
IL-6-IL-17	r = −0.3748 *p* = 0.2299	r = −0.1555 *p* = 0.5800
IL-10-IL-17	r = −0.1863 *p* = 0.5621	r = −0.0876 *p* = 0.7563

Note: MLT—melatonin. IL—Interleukin. r = Pearson coefficient.

**Table 4 ijms-24-08647-t004:** Daily Living Activities, Basic Activities by Katz Index, and Instrumental Activities by Lawton and Brody.

Comprehensive Geriatric Assessment	Control % (N)	Infected % (N)
Shower Alone	97.5 (39)	100 (20)
Dress Alone	97.5 (39)	95.0 (19)
Unassisted toilet use	97.5 (39)	95.0 (19)
Mobility (without help)	95.0 (38)	95.0 (19)
Feeding (without help)	97.5 (39)	95.0 (19)
Using the phone without assistance	90.0 (36)	85.0 (17)
Travel alone	82.5 (33)	80.0 (16)
Shopping	75.0 (30)	80.0 (16)
Meal preparation	70.0 (28)	35.0 (7)
Ability to perform heavy tasks		
Incapable/has no habit	27.5 (11)	35.0 (7)
Light	22.5 (9)	50.0 (10)
Heavy	47.5 (19)	60.0 (12)
Use of medication without help	82.5 (33)	90.0 (18)
Ability to count money	95.0 (38)	80.0 (16)

Legend: N = number; % = percentage.

## Data Availability

The authors will make the data supporting this study’s interpretations available if requested.

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
