# Peer review of "Melatonin and Cytokines Modulate Daily Instrumental Activities of Elderly People with SARS-CoV-2 Infection"

_ijms, 2023, doi:10.3390/ijms24108647_

Round 1

Reviewer 1 Report

This manuscript clarified changes in melatonin and cytokines concentrations in the elderly people with Sars-Cov-2 infections and provides important information of immune responses in the elderly people.

There are some questions as follows.

1. Efficacy of comprehensive geriatric assessment (CGA) is different among environments of participants, home geriatric assessment, acute geriatric care units, post-hospital discharge, outpatient consultation, inpatients consultation. Participants environments were all same or not?

2. In obesity and diabetes, blood melatonin levels change variously (Int J Mol Sci. 2023, 24:3300; Curr Diabetes Rev. 2021, 17:e072620184137; Biology, 2020, 7:72). Are there any participants with obesity or diabetes in this study?

3. In the infected participants, serum melatonin concentrations increased significantly. The mechanism of increasing of serum melatonin should be mentioned.

4. Melatonin has been noted for its anti-inflammatory, anti-oxidative, anti-apoptotic, and neuroprotective actions (J Inorganic Biochem. 2021, 223:111546). What is the role of increased melatonin in the infected participants?

Author Response

Thank you very much for the suggestions for improving the manuscript. All changes have been made and are described below.

( ) I would not like to sign my review report

(x) I would like to sign my review report

Quality of English Language

(x) I am not qualified to assess the quality of English in this paper

( ) English is very difficult to understand/incomprehensible

( ) Extensive editing of the English language required

( ) Moderate editing of the English language

( ) Minor editing of the English language required

( ) English language is fine. No issues detected

Yes

Can be improved

Must be improved

Not applicable

Does the introduction provide sufficient background and include all relevant references?

(x)

( )

( )

( )

Are all the cited references relevant to the research?

(x)

( )

( )

( )

Is the research design appropriate?

( )

(x)

( )

( )

Are the methods adequately described?

( )

(x)

( )

( )

Are the results clearly presented?

(x)

( )

( )

( )

Are the conclusions supported by the results?

(x)

( )

( )

( )

Comments and Suggestions for Authors

This manuscript clarified changes in melatonin and cytokines concentrations in elderly people with Sars-Cov-2 infections and provided important information on immune responses in elderly people.

There are some questions as follows.

  1. The efficacy of comprehensive geriatric assessment (CGA) is different among environments of participants, geriatric home assessment, acute geriatric care units, post-hospital discharge, outpatient consultation, and inpatient consultation. Participants environments were all the same or not?

R: All elderly people evaluated in this study lived at home with family members or caregivers. This information was added in Table 1.

  1. In obesity and diabetes, blood melatonin levels change variously (Int J Mol Sci. 2023, 24:3300; Curr Diabetes Rev. 2021, 17:e072620184137; Biology, 2020, 7:72). Are there any participants with obesity or diabetes in this study?

R: Of the 116 patients evaluated, 28 were excluded for having diabetes and/or obesity, and 15 patients aged below 60 years. This information was added to materials and methods.

  1. In the infected participants, serum melatonin concentrations increased significantly. The mechanism of increasing serum melatonin should be mentioned.

R: One of the main actions of melatonin is to act as an anti-inflammatory. As in patients with COVID-19, there was an increase in pro-inflammatory cytokines; the increase in the hormone melatonin probably occurred due to a response to the inflammatory environment as a feedback mechanism for controlling inflammation during infection, for the re-establishment of homeostatic processes. This information has been improved in the conclusions.

  1. Melatonin has been noted for its anti-inflammatory, anti-oxidative, anti-apoptotic, and neuroprotective actions (J Inorganic Biochem. 2021, 223:111546). What is the role of increased melatonin in the infected participants?

R: In patients with COVID-19, the role of melatonin may be related to the control of inflammation since there was a correlation between the increase in this hormone and the increase in IL-6 and IL-10.

Reviewer 2 Report

The authors measure too many things in too few patients. No sample size calculation or attained power are given.

The conclusion has to be toned down

More or less ok

Author Response

Thank you very much for the suggestions for improving the manuscript. All changes have been made and are described below.

(x) I would not like to sign my review report

( ) I would like to sign my review report

Quality of English Language

( ) I am not qualified to assess the quality of English in this paper

( ) English is very difficult to understand/incomprehensible

( ) Extensive editing of the English language required

( ) Moderate editing of the English language

(x) Minor editing of English language required

( ) English language is fine. No issues detected

Yes

Can be improved

Must be improved

Not applicable

Does the introduction provide sufficient background and include all relevant references?

( )

(x)

( )

( )

Are all the cited references relevant to the research?

(x)

( )

( )

( )

Is the research design appropriate?

( )

( )

(x)

( )

Are the methods adequately described?

(x)

( )

( )

( )

Are the results clearly presented?

( )

(x)

( )

( )

Are the conclusions supported by the results?

( )

( )

(x)

( )

Comments and Suggestions for Authors

The authors measure too many things in too few patients. No sample size calculation or attained power are given.

R: The sample calculation showed that a sample of 28 individuals, in each group would allow a significance level of 95% and a sampling power of 97.2%. This information was added in Materials and Methods section

The conclusion has to be toned down

R: The conclusion was improved.

Comments on the Quality of English Language

More or less ok

R:  Thank you for the suggestions, and we apologize for any inconvenience. The text was improved and revised by a native English speaker.

Round 2

Reviewer 1 Report

This manuscript is revised well according to the reviewers' comments.

This manuscript provides important information of immune responses in elderly people.